# A Rapid Method for the Identification of Fresh and Processed *Pagellus erythrinus* Species against Frauds

**DOI:** 10.3390/foods9101397

**Published:** 2020-10-02

**Authors:** Marina Ceruso, Celestina Mascolo, Pasquale De Luca, Iolanda Venuti, Giorgio Smaldone, Elio Biffali, Aniello Anastasio, Tiziana Pepe, Paolo Sordino

**Affiliations:** 1Department of Veterinary Medicine and Animal Production, University of Naples Federico II, via F. Delpino, n.1, 80137 Naples, Italy; marina.ceruso@gmail.com (M.C.); celeste.mascolo@gmail.com (C.M.); iolandavenuti@gmail.com (I.V.); anastasi@unina.it (A.A.); 2Department of Biology and Evolution of Marine Organisms, Stazione Zoologica Anton Dohrn, Villa Comunale, 80121 Naples, Italy; paolo.sordino@szn.it; 3Department of Research Infrastructures for Marine Biological Resources, Stazione Zoologica Anton Dohrn, Villa Comunale, 80121 Naples, Italy; pasquale.deluca@szn.it (P.D.L.); elio.biffali@szn.it (E.B.); 4Department Agricultural Sciences, University of Naples Federico II, via Università, n.100, Portici, 80055 Naples, Italy; giorgio.smaldone@unina.it

**Keywords:** fish species authentication, mtDNA, mitogenomics, common pandora, *Pagellus erythrinus*, *Sparidae*

## Abstract

The commercialization of porgies or seabreams of the family *Sparidae* has greatly increased in the last decade, and some valuable species have become subject to seafood substitution. DNA regions currently used for fish species identification in fresh and processed products belong to the mitochondrial (mt) genes cytochrome b (*Cytb*), cytochrome c oxidase I (*COI*), *16S* and *12S*. However, these markers amplify for fragments with lower divergence within and between some species, failing to provide informative barcodes. We adopted comparative mitogenomics, through the analysis of complete mtDNA sequences, as a compatible approach toward studying new barcoding markers. The intent is to develop a specific and rapid assay for the identification of the common pandora *Pagellus erythrinus*, a sparid species frequently subject to fraudulent replacement. The genetic diversity analysis (Hamming distance, *p*-genetic distance, gene-by-gene sequence variability) between 16 sparid *mt*DNA genomes highlighted the discriminating potential of a 291 bp *NAD2* gene fragment. A pair of species-specific primers were successfully designed and tested by end-point and real-time PCR, achieving amplification only in *P. erythrinus* among several fish species. The use of the *NAD2* barcoding marker provides a rapid presence/absence method for the identification of *P. erythrinus*.

## 1. Introduction

Food frauds are considered a major safety, quality, and economic problem worldwide, with rising awareness and concern in consumers [1,2,3,4,5]. In Europe, mislabeling is responsible for 41.8% of food fraud violations and fish products have a central role in this scenario [6]. The International Food Safety Authorities Network (INFOSAN) recently pointed out that there is a common desire from all countries for more technical support regarding food frauds prevention and management [1]. Fish and fish products are the most subject to fraud in the European Union, with intentional mislabeling being the main type of violation [6]. The common pandora (*Pagellus erythrinus* Linnaeus, 1758) is one of the most commercially caught *Sparidae* species in the Mediterranean Sea and the Atlantic Ocean. The commercialization of *P. erythrinus*, deriving from fishing activity between 2006 and 2016, has expanded from 9.34 to 13.90 tonnes (+48.72%), while the aquaculture production has progressively dropped from 197.00 tonnes in 2006 to 0.04 in 2016 [7]. Most of the wild common pandora catches come from Mediterranean fishery grounds (12.25 tonnes), mainly in Italy, Libya, Spain, and Tunisia, with a small share from Atlantic countries (1.65 tonnes) such as Morocco [7]. Fish farm production of *P. erythrinus* was 127.27 tonnes in 2014 in Greece, and 0.23 tonnes in 2016 in Cyprus [8]. Owing to the increasing appreciation of European consumers, *P. erythrinus* is deplorably subject to replacement with less valuable species. Fraudulent substitution of the common pandora occurs not only when prepared and processed specimens have lost external characters but also as whole fish, due to the morphological resemblance with other sparid species (e.g., *Pagellus acarne*, *Pagellus bellottii*, *Pagrus pagrus*, *Lutjanidae* spp.) [9,10,11,12,13]. Currently, cytochrome b-*Cytb*, cytochrome c oxidase I-*COI*, *16S*, and *12S* genes of the mitochondrial (mt) DNA are traditionally used for fish species identification [14,15,16]. However, these universal barcodes are not always effective for unambiguous species authentication as well as for forensic approaches such as fraud detection [14,17,18]. The limitation of providing adequate resolution at deep nodes emerges especially when fish species have a high degree of genetic homology. Traditional markers are inadequate, especially when they amplify for fragments that differentiate species by point mutations [19]. We have recently shown that the analysis of the complete mtDNA (mitogenomics) provides a useful approach for identifying fast and adequate barcoding markers. In particular, our previous study showed that the *NAD5* gene, compared to traditional markers, possesses a higher discrimination capacity for species of the family Sparidae [20]. *NAD5* is coding for the subunit 5 of NADH dehydrogenase, part of the mitochondrial membrane, involved in the electron transport to the respiratory chain [21]. This gene consists of short variable sequence regions associated with conserved areas that allowed to obtain universal primers for sparids, amplifying in all species for a 265 bp fragment.

Here, a more detailed study on the genetic distance between the complete mtDNA of species belonging to the family *Sparidae* was carried on using a gene-by-gene approach. The updated mtDNA comparison allowed identifying *NAD2* as a new barcoding gene that agrees to discriminate species within the same genus, based on its extremely high level of divergence in the nucleotide sequence. Our applicative research shifted toward the development of presence/absence assay for discriminating *P. erythrinus*. In this study, we propose a *NAD2* fragment as a specific marker for the common pandora. The gene *NAD2* was previously investigated as a marker for population studies of plants, parasites and animals [22,23,24,25]. However, no use is made of this gene, as far as we know, in fish species identification against frauds, and our results show that its use warrants further investigation. We designed species-specific primers, testing their performance in both classical and Real-Time PCR. The present research allows and simplifies common pandora authentication since a classical PCR sequencing-free or RT-PCR without electrophoresis concludes analysis and significantly reduces the time and costs needed for correct species identification.

## 2. Materials and Methods

### 2.1. Fish Samples

Ten common pandora specimens from different FAO areas (27, 34, and 37) were sampled and used to test the PCR primer species-specificity (Table 1, Figure 1). The geographical coordinates of the fishing spots were provided by fish market operators. In fact, for the traceability protocol established by EC Reg. 1224/2009, each fishing company with boats over 10 m must have an electronic logbook for detecting the point where the nets are lowered (trawl fishing).

In the FAO area 37, spanning most of the species distribution (www.aquamaps.org), eight specimens were harvested from as many Geographical Sub-Areas (GSA) (Figure 1). The evaluation of primer specificity was extended to the other 26 fish species, as reported in Table 2.

These additional species were selected for the purpose to comprise (i) those used to replace *P. erythrinus* (e.g., *Pagellus acarne*), (ii) phylogenetically related sparid species (e.g., *Dentex dentex*) [26], and (iii) other species commonly found in Mediterranean fish markets. Eleven specimens of *D. dentex*, the most genetically related sparid species (Figure 2), were sampled, as reported in Table 3.

All additional species were provided by Pozzuoli (Naples, Italy) and Salerno (Italy) fish markets. Specimens were labeled and conserved on board at −20 °C, transported in isothermal conditions to fish markets, and then to the research laboratory. The classification at the species level was carried out on the bases of fishes anatomical and morphological characteristics at the Department of Veterinary Medicine and Animal Production, University of Naples Federico II, Naples (Italy).

### 2.2. Total Genomic DNA Extraction

Total DNA extraction for each sample was performed in double from muscle tissue using the DNeasy Blood & Tissue Kit (Qiagen, Hilden, Germany) according to the procedure proposed by the manufacturer [27]. DNA concentration was determined with Nanodrop 2000 spectrophotometer (Thermo Fisher Scientific, Waltham, MA, USA). The range of DNA amount was 40 ng/µL, while the purity was 1.8–2.0 ratio at A260/A280. DNA quantity and quality were evaluated by electrophoretic analysis in 1% agarose gel.

### 2.3. Comparative Analysis of mtDNA Complete Sequences

Sixteen complete mitogenome sequences of sparid species were analyzed. Ten of the sixteen were available in GenBank and six were sequenced and deposited in NCBI by this research group (Table 4). MtDNA sequences were analyzed and compared by the use of several bioinformatics tools, with the aim to find species-specific gene fragments for *P. erythrinus* identification. Unipro UGENE software [28] was used to perform the alignment. With the aim to evaluate the genetic divergence among genes and species, the hamming distance algorithm was used [29]. The *p-*genetic distance within species, the nucleotide sequence variability, pairwise and multiple alignments and gene divergence were determined as previously described [20,30,31].

### 2.4. NAD2 Fragment Amplification and Sequence Analysis

A pair of *P. erythrinus*-specific *NAD2* primers that amplified a 291 bp fragment was designed by eye after multiple alignments of the 16 sparid species complete mtDNA sequences using BioEdit Sequence Alignment Editor [46]. In particular, primer design was carried out so that the nucleotide base at 3′ end was species-specific. Melting temperature (Tm), secondary structure, self-annealing, and inter-primer binding, were verified using Multiple Primer Analyzer (Thermo Fisher Scientific, Waltham, MA, USA). *NAD2* primers efficiency for sparid species identification was further verified in silico [28].

PCR amplifications were performed as reported in Ceruso et al., 2019 [20], with the annealing temperature at 57 °C (291 bp) for 1 min. PCR products were purified using the QIAquick PCR Purification Kit (Qiagen). PCR reactions were carried out on mtDNA extracted from 10 fresh *P. erythrinus* specimens originating from different GSA/FAO areas (Table 1, Figure 1) and fresh specimens belonging to several other species of commercial interest (Table 2 and Table 3).

With the aim to confirm the correct amplification of the *NAD2* gene fragment, amplicons sequences were assessed with the Sanger method using the Automated Capillary Electrophoresis Sequencer 3730 DNA Analyzer (Applied Biosystems, Foster City, CA, USA) at the Molecular Biology Service at the Stazione Zoologica Anton Dohrn. *NAD2* sequences were evaluated with the BioEdit Sequence Alignment Editor. The concordance between morphological and molecular analyses was assessed running a BLAST analysis of the obtained sequences on GenBank for species identification [47].

### 2.5. Real-Time PCR

To assess the presence or absence of an amplicon with a simple yes/no answer, making it similar to a terminal PCR and gel electrophoresis, we tested primer sensibility and specificity in a protocol for Real-Time PCR (RT-PCR) technique. For some labs, it could be useful to assemble a reaction, load it into a single instrument, and obtain the needed information by visualizing species-specific amplification without the additional electrophoresis step. The DNA samples were used as a template for RT-PCR experiments performed in a Viia7 real-time PCR system (Applied Biosystems) at 1:10 dilution, with the primers described before [48]. The PCR volume of each sample was 10 μL with 5 μL of 2× RT-PCR SYBR Green Master Mix (Thermo Fisher), 0.05 or 0.1 pmol/μL for each primer and 2 μL of 1:10 diluted DNA template. The expected fragment of 291 bp in length had a theoretical melting temperature of about 87 °C calculated by Endmemo (http://www.endmemo.com/bio/tm.php). A slight difference in RT-PCR experiments is expected, due to the specific instrument used and the reaction conditions. In the set-up experiment, a specific amplicon with a melting temperature of about 80 °C was detected (data not shown), so we decided to test all fish specimens. Experiments were performed in triplicate. The diagram was elaborated with Excel.

## 3. Results

### 3.1. Pagellus erythrinus mtDNA Comparative Data

Hamming distance comparison results showed that the genetic dissimilarity among mtDNA of *P. erythrinus* and other sparid species varies from a minimum of 10% (*Dentex dentex*) to a maximum of 19% (*Acanthopagrus latus* and *Rhabdosargus sarba*) (Figure 2). In Figure 2, the species are ordered clockwise from the species closest to the farthest from *P. erythrinus*. Statistical analysis was reported in Ceruso et al., 2019 [20].

The Hamming distance was also studied comparing each mitochondrial gene. Results showed that the five genes with higher values of genetic dissimilarity were *ATP6* (25%), *NAD2* (24%), *NAD4* (24%), *NAD6* (24%), and *NAD5* (23%) (Figure 3).

The *p*-genetic distance analysis among all sparids mtDNA displayed that the *NAD* group genes have a sequence distance ranging between 0.20 and 0.25, more than *COI* (0.16) and *Cytb* genes (0.19) (Figure 4). 

Accordingly, nucleotide sequence variability analysis displayed the highest values in the NAD gene group (NAD1 39%, NAD2 50%, NAD3 39%, NAD4L 39%, NAD4 43%, NAD5 41%, NAD6 44%) (Figure 5). About amino acidic sequences, NAD2 (35%) and ATP8 (32%) proteins showed the highest variability (Figure 5).

### 3.2. NAD2 Amplification and Analysis

Following the results of Sparids mitogenome comparison, primers for amplifying species-specific nucleotide sequences were designed on the NAD2 gene. The high degree of nucleotide sequence variability of this gene among sparids species allowed to correctly design primers amplifying a 291 bp (from 303 to 593 nt) fragment so the nucleotide base at 3′ end was species-specific for *P. erythrinus* (Table 5, Figure 6).

As reported in the Materials and Methods Section 2.4, the specificity of the designed primers was firstly tested in silico against all available Sparidae mitogenomes (Table 4).

Endpoint-PCR amplification was obtained in all the 10 genomic DNA samples of *P. erythrinus*. No amplification occurred in the DNAs of other fish species (Figure 7).

The ten *P. erythrinus* amplicons were sequenced to confirm the correct amplification of the 291 bp NAD2 fragment. Amplicon sequence comparison with databases showed accurate species identification, with similarity scores of NAD2 sequences ranging between 98% and 100%.

The value of intraspecific genetic variability ranged between 0% and 0.8%, with three different nucleotides found in two out of ten specimens (specimens Pe9 and Pe10 in Table 1).

### 3.3. NAD2 Amplification by RT-PCR

All the *P. erythrinus* samples gave an amplification with a Ct (Cycle Threshold) between 23 and 31 (a low Ct indicates the presence of the specific fragment used in the experiment) and a Tm between 78 °C and 79 °C (as expected). All the other species gave an amplification with Ct between 36 and 40 (where 40 means undetectable signal, like the negative control) and Tm that varied between 60 °C (the negative control) and 72 °C (Figure 8), indicating the presence of a very small amount of a non-specific product.

## 4. Discussion

Current studies focused on DNA based fish species identification use mitochondrial markers (e.g., *COI*, *Cytb*, *12S* and *16S*) standardly, assuming that they have discriminating ability on all fish families, genus or species. Nevertheless, the poor reliability of most widely used mitochondrial genes for barcoding and for constructing phylogenetic trees has been outlined earlier [49,50,51,52]. Thus, the high degree of genetic homology among fish species of the same family may cause limitations to providing adequate resolution for a correct identification, since species are differentiated by point mutations. Currently, mitochondrial genomes can be rapidly obtained from genome or transcriptome datasets [27], but comparative mitogenomics has been barely used for the discovery of new specific genes useful as markers for species and strain recognition [19,53,54].

The first innovation of this paper is that a vast amount of data was explored, considering the complete mtDNA of sixteen *Sparidae* species, to verify the presence of genes and gene fragments with more genetic divergence than standard markers. Thus, the investigation of the complete mitochondrial genome of the most commercially important sparid species allowed us to identify a new barcoding marker. The new approach showed that the “top five genes” with higher values of genetic dissimilarity among sparids were 1-*ATP6*, 2-*NAD2*, 3-*NAD4*, 4-*NAD6*, and 5-*NAD5*. It is important to note that among these five genes, there are two (*ATP6*-684 bp and *NAD6*-522 bp) with a total length of much less than others (*NAD2*-1047 bp, *NAD4*-1381 bp and *NAD5*-1839 bp). In the three longer genes, the genetic dissimilarity degree appears to have greater statistical significance. Another consideration is that this ranking does not include genes currently used for sparid species identification (*COI*, *Cytb*, *12S* and *16S*).

The *p*-genetic distance analysis confirmed previous results, with the same “top five genes”, but in a different order: 1-*NAD2*; 2-*NAD5*; 3-*NAD4*; 4-*ATP6* and 5-*NAD6.*

Lastly, also the gene sequence variability study completed and reinforced previous findings. The best genes were 1-*NAD2,* followed by 2-*ATP6*; 3-*NAD6,* 4-*NAD4* and 5-*ATP8.* Once again, *NAD* genes group have higher values compared with traditional barcoding genes used for sparid species identification, i.e., *Cytb*, *COI*, *16S* and *12S*.

Another important consideration and innovation of our study is that the comparison carried out allowed to find a new gene, never considered before for sparids barcoding scopes (*NAD2*), with an incisive, rapid, and unequivocal discriminating ability. Therefore, applying bioinformatics analysis, the barcoding value of all the *NAD* group genes was previously reported in fish families, but the *NAD5* gene was selected as a potential marker [20,55].

One of the main features that differentiates the *NAD2* gene from the *NAD5* is the distribution of genetic diversity. In fact, the *NAD5* had highly divergent areas in alternance with very similar areas between species. This made it possible to design primers that amplified across all species, allowing diversification through sequencing only. Instead, the *NAD2* high degree of genetic divergence is spread throughout all the gene, in order to allow the design of primers with a 3′ primer start position different from the other species hence specific for *P. erythrinus*. Clearly, this *NAD*2 feature makes it possible to design species-specific primers for other sparid species, bringing great benefits in the research field of species identification against frauds.

Finally, this research provides an expansion of knowledge on the genetic dissimilarity among *Sparids*. Previous research on the genetic distance among sparid species in the family *Sparidae* focalized on single genes (e.g., *Cytb)*, not on the complete mtDNA [56].

The new approach based on the complete mtDNA genome has allowed us to know the potential information content on genetic variation and lineage divergence of *Sparidae* species. The comparative mitogenomic analysis reveals the close relationship of *P. erythrinus* with other sparid species, with genetic divergence ranging from 10 to 19%. Gene-by-gene Hamming distance analysis identifies *ATP6*, *NAD2*, *NAD4*, *NAD5*, and *NAD6* as the less conserved mitochondrial genes among porgies and seabreams.

Our results allowed to find the species-specific barcode marker for *P. erythrinus*, to use in frauds prevention based on species substitution. This research represents a new way to approach the studies on fish species identification and should be applied to other fish species to prevent and manage frauds.

## 5. Conclusions

An in-depth analysis of the complete mitogenomes of sparid fishes provides a novel species-specific barcoding marker for the identification of the common pandora *P. erythrinus*, a seafood industry product commonly subject to frauds by replacement. This study reports a consistent and fast presence/absence visual assay to prevent frauds by substitution of *P. erythrinus*, through a simple end-point PCR that does not need sequencing. The RT-PCR confirmed primer specificity and sensibility and does not require electrophoresis but implies more laboratory skills. 

The Official Controls Regulation (EU) 2017/625, gives increasing importance to consumer protection and safety against fraudulent practices along the agri-food chain. The replacement and mislabeling of fish species with others of lower commercially value is a growing problem in the production and distribution of fishery products chain. Effective species identification in fresh, frozen, and treated products may contribute to the “molecular traceability” of seafood, in agreement with Regulation (EU) 1379/2013 (European Commission, Brussels, Belgium, 2013). Competent national authorities could make the food control and authentication activities more effective thanks to the full use of DNA test analysis, in order to discourage fraudulent species replacement activities.

## Figures and Tables

**Figure 1 foods-09-01397-f001:**
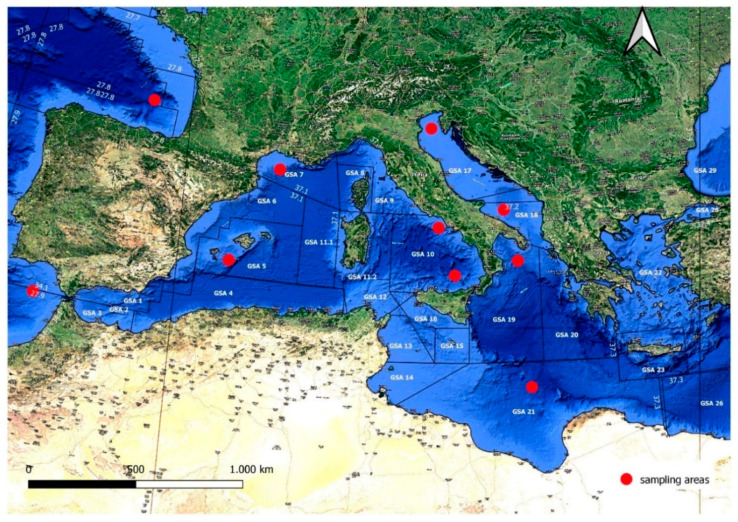
Red spots show the geographical origins of the *P. erythrinus* specimens sampled for this study. The sampling areas considered the GSA division from GFCM (Establishment of Geographical Sub-Areas in the GFCM area amending the resolution GFCM/31/2007/2) and FAO areas. GFCM: General Fisheries Commission for the Mediterranean. GSA: Geographical Sub-Area.

**Figure 2 foods-09-01397-f002:**
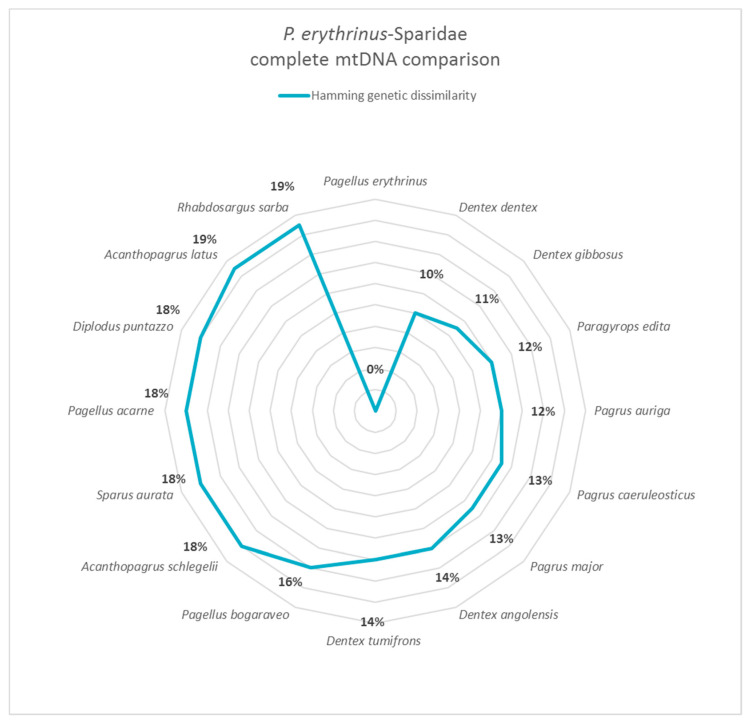
Hamming distance comparison showed that the genetic dissimilarity among mtDNA of *Pagellus erythrinus* and other sparid species ranges from 10% (*Dentex dentex)* to 19% (*Acanthopagrus latus* and *Rhabdosargus sarba*).

**Figure 3 foods-09-01397-f003:**
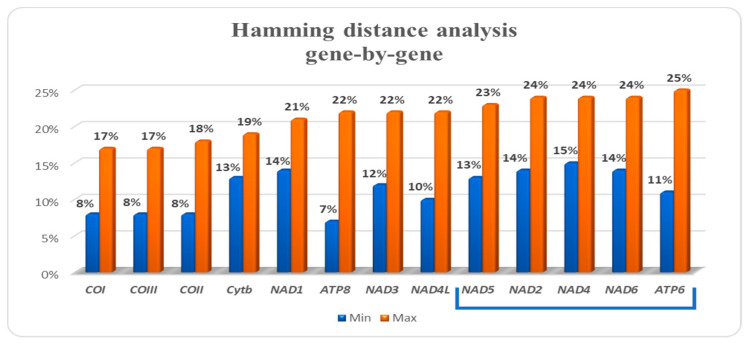
Gene-by-gene Hamming distance analysis carried out between *P. erythrinus* and other sparid species. The blue bracket shows the five less conserved mitochondrial genes in the *Sparidae* family.

**Figure 4 foods-09-01397-f004:**
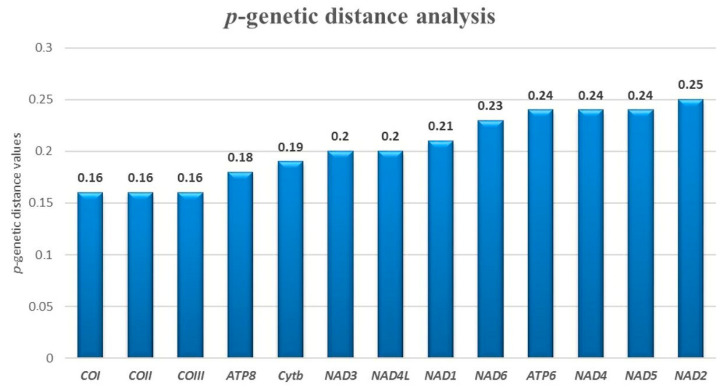
*p*-genetic distance analysis. The genes were sorted by increasing *p*-distance values. The gene with the lowest value is *COI*, while the gene with the highest value is *NAD2*.

**Figure 5 foods-09-01397-f005:**
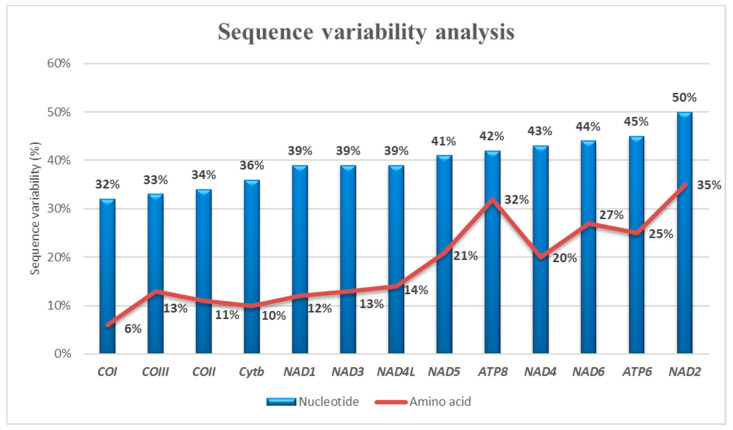
Gene-by-gene analysis of nucleotide and amino acid sequence interspecific variability. The genes were sorted by increasing variability values. The gene with the lowest value is *COI*, while the gene with the highest value is *NAD2.*

**Figure 6 foods-09-01397-f006:**
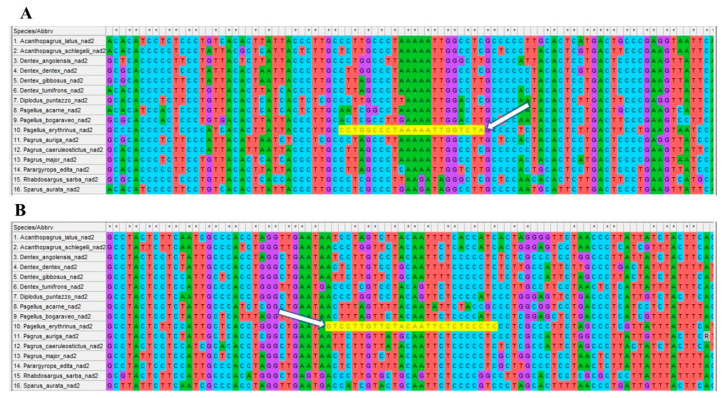
Species-specific primer alignment on Sparidae *NAD2* orthologous genes. Primers (highlighted in yellow) aligned only in *P. erythrinus* (sample n. 10). White arrows show the 3′ end species-specific position. (**A**) Primer FW303. (**B**) Primer REW593 (reverse-complement).

**Figure 7 foods-09-01397-f007:**
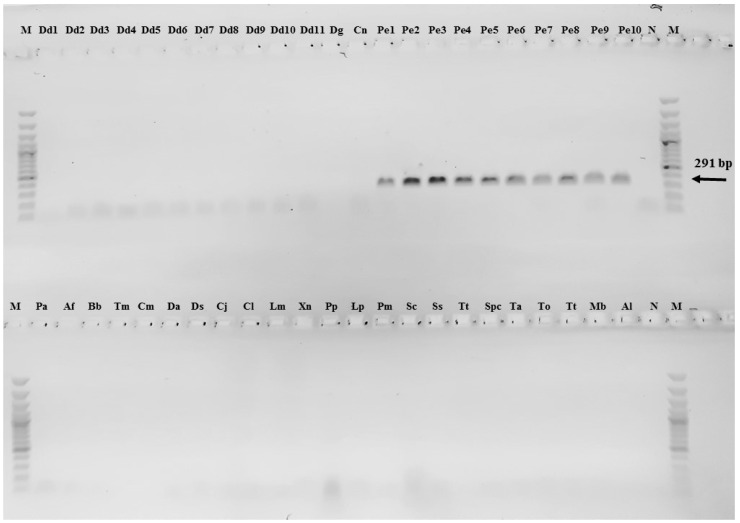
Gel electrophoretic image. End-point PCR amplification of *NAD2* fragment in ten geographically disjunct specimens of *P. erythrinus* (lanes 14–35, from Pe1 to Pe10). No amplification was obtained for all other fish species (lanes 1–13 and 36–46). Abbreviations as in Table 1, Table 2 and Table 3. N: Negative control; M: 100 bp ladder.

**Figure 8 foods-09-01397-f008:**
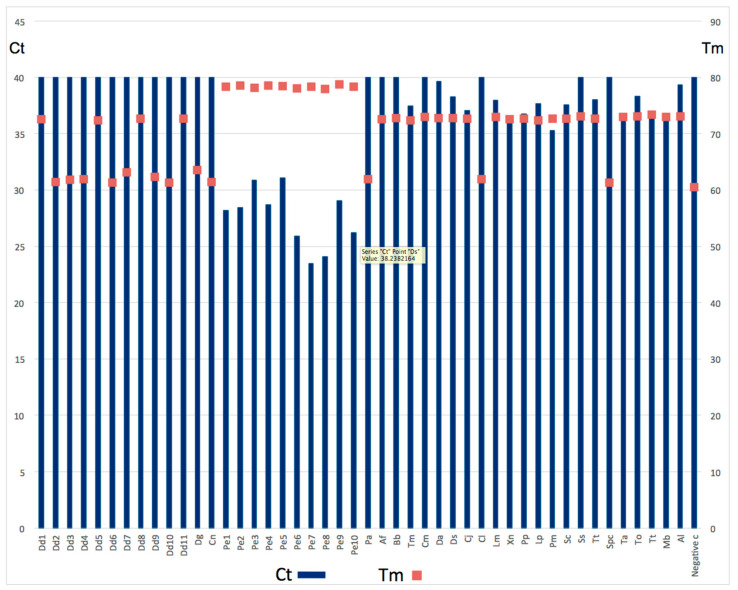
Real-Time PCR. Threshold cycles (Ct) on the left and melting temperature (Tm) on the right. Abbreviations as in Table 2, Table 3 and Table 4.

**Table 1 foods-09-01397-t001:** Geographical origins of the *P. erythrinus* specimens sampled for this study.

*P. erythrinus*			
Abbreviation	Latitude	Longitude	Sampling Areas
Pe1	38.714589	2.230898	FAO 37 area—GSA5—Balearic Island
Pe2	42.774369	4.384218	FAO 37 area—GSA7—Gulf of Lions
Pe3	40.775480	13.603978	FAO 37 area—GSA10—South Tyrrhenian
Pe4	38.748871	14.524641	FAO 37 area—GSA10—Central Tyrrhenian
Pe5	44.991156	13.065691	FAO 37 area—GSA17—Northern Adriatic
Pe6	41.533674	17.278337	FAO 37 area—GSA18—Southern Adriatic Sea
Pe7	39.343228	17.981462	FAO 37 area—GSA19—Western Ionian Sea
Pe8	33.952939	18.508806	FAO 37 area—GSA21—Southern Ionian Sea
Pe9	44.708601	−3.470325	FAO 27 area—Atlantic Ocean
Pe10	35.916767	−7.425403	FAO 34 area—Atlantic Ocean

**Table 2 foods-09-01397-t002:** Fish species other than *P. erythrinus* used in this study by end-point PCR and Real-Time qPCR. Common names are from ASFIS List of Species for Fishery Statistics Purposes (http://www.fao.org/fishery/collection/asfis/en).

Scientific Name	Family	Common Name	Abbreviation
*Dentex dentex*	*Sparidae*	Common dentex	Dd
*Dentex gibbosus*	*Sparidae*	Pink dentex	Dg
*Cheimerius nufar*	*Sparidae*	Santer seabream	Cn
*Pagellus acarne*	*Sparidae*	Axillary seabream	Pa
*Aulopus filamentosus*	*Aulopidae*	Royal flagfin	Af
*Boops boops*	*Sparidae*	Bogue	Bb
*Trisopterus minutus*	*Gadidae*	Poor cod	Tm
*Cepola macrophthalma*	*Cepolidae*	Red bandfish	Cm
*Diplodus annularis*	*Sparidae*	Annular seabream	Da
*Diplodus sargus*	*Sparidae*	White seabream	Ds
*Coris julis*	*Labridae*	Rainbow wrasse	Cj
*Chelidonichthys lucerna*	*Triglidae*	Tub gurnard	Cl
*Lithognathus mormyrus*	*Sparidae*	Sand steenbras	Lm
*Xyrichtys novacula*	*Labridae*	Pearly razorfish	Xn
*Pleuronectes platessa*	*Pleuronectidae*	European plaice	Pp
*Lophius piscatorius*	*Lophiidae*	Angler (= Monk)	Lp
*Scophthalmus maximus*	*Scophthalmidae*	Turbot	Pm
*Sebastes capensis*	*Sebastidae*	Cape redfish	Sc
*Solea solea*	*Soleidae*	Common sole	Ss
*Trachurus trachurus*	*Carangidae*	Atlantic horse mackerel	Tt
*Spondyliosoma cantharus*	*Sparidae*	Black seabream	Spc
*Thunnus albacares*	*Scombridae*	Yellowfin tuna	Ta
*Thunnus obesus*	*Scombridae*	Bigeye tuna	To
*Thunnus thynnus*	*Scombridae*	Atlantic bluefin tuna	Tt
*Mullus barbatus*	*Mullidae*	Red mullet	Mb
*Arnoglossus laterna*	*Bothidae*	Mediterranean scaldfish	Al

**Table 3 foods-09-01397-t003:** Geographical origins of the *D. dentex* specimens sampled for this study.

*Dentex Dentex*			
Abbreviation	Latitude	Longitude	Sampling Areas
Dd1	39.808662	3.740905	GSA 5—Balearic Island
Dd2	42.519035	3.534141	GSA 7—Gulf of Lions
Dd3	38.771348	14.946960	GSA 10—South Tyrrhenian
Dd4	40.409446	13.798889	GSA 10—Central Tyrrhenian
Dd5	40.044575	9.841622	GSA 11.2—Sardinia (East)
Dd6	44.980473	13.137520	GSA 17—Northern Adriatic
Dd7	40.668284	18.290108	GSA 18—Southern Adriatic Sea
Dd8	40.044587	17.147530	GSA 19—Western Ionian Sea
Dd9	31.475938	18.685467	GSA 21—Southern Ionian Sea
Dd10	35.338496	35.708826	GSA 27—Levante
Dd11	40.874620	12.985808	GSA 10—Ponza Island, Central Thyrrhenian

**Table 4 foods-09-01397-t004:** Sparids species with whole mtDNA sequences available in GenBank evaluated in this research.

Species	Ac. Number	FAO Fishing Areas	References
*Acanthopagrus latus*	NC_010977	FAO 71	[32]
*Acanthopagrus schlegelii*	JQ_746035	FAO 71	[33]
*Dentex angolensis*	NC_044097	FAO 47	[34]
*Dentex dentex*	MG_727892	FAO 37	[35]
*Dentex gibbosus*	MG_653593	FAO 37	[36]
*Dentex tumifrons*	NC_029479	FAO 71	[37]
*Diplodus puntazzo*	MT319027	FAO 37	[26]
*Pagellus acarne*	MG_736083	FAO 37	[38]
*Pagellus bogaraveo*	NC_009502	FAO 27	[39]
*Pagellus erythrinus*	MG_653592	FAO 37	[40]
*Pagrus auriga*	AB_124801	FAO 37	[41]
*Pagrus caeruleostictus*	MN319701	FAO 34	[42]
*Pagrus major*	NC_003196	Farmed and supplied from Andalusia (Spain) fish market	[43]
*Parargyrops edita*	EF_107158	FAO 71	[44]
*Rhabdosargus sarba*	KM_272585	Farmed Daya Bay Aquaculture Center, Guangdong (China)	[18]
*Sparus aurata*	LK_022698	Farmed and supplied from Jaffa (Israel) fish market	[45]

**Table 5 foods-09-01397-t005:** *P. erythrinus* species-specific *NAD2* primers.

Primer Name	5′→3′ Sequence	Tm °C	CG%	nt	A	T	C	G
FW303	CCTGGCCCTAAAAATTGGTCTA	65.1	45.5	22	6	6	6	4
REV593	GAGGGAGAGAATTGTAGAACAAGGAC	65.4	46.2	26	11	3	2	10

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
