# Peer review of "A Rapid Method for the Identification of Fresh and Processed Pagellus erythrinus Species against Frauds"

_foods, 2020, doi:10.3390/foods9101397_

Round 1

Reviewer 1 Report

I have some editorial comments as below.

Abstract

L. 18: seabreams (Sparidae)→seabreams of the family Sparidae

L. 21: a Sparidae species→a sparid species

L. 24: 16 Sparidae→16 sparid

Introduction

L 47-48: mitochondrial (mt) DNA regions of the cytochrome b-Cytb, cytochrome c oxidase I-COI, 16S, and 12S genes→cytochrome b-Cytb, cytochrome c oxidase I-COI, 16S, and 12S genes of the mitochondrial (mt) DNA

L. 55: NAD2?

L. 56: for Sparidae→for species of the family Sparidae

L. 57: fragments→regions

L. 59: mitochondria→mtDNA

L. 60: Sparidae family→family Sparidae

L. 61: mitochondrial→mtDNA

Materials and Methods

L. 105: MtDNA

L. 115: 16 sparid species

Results

140: spell out P. erythrinus

p of p-genetic distance is in italic or not?

Table and Figure caption

Spell out all generic name.

L. 198: What is Ct? This seems to show up first here, but no explanation can be seen anywhere except for the caption of Fig. 8.

Discussion

L. 223, 230, 236: sparid

L. 249: sparid species in the family Sparidae

L. 257: P. erythrinus

Figure 2. other sparid species

Author Response

-     Abstract

  1. 18: seabreams (Sparidae)→seabreams of the family Sparidae DONE
  2. 21: a Sparidae species→a sparid species DONE
  3. 24: 16 Sparidae→16 sparid DONE
  • Introduction

L 47-48: mitochondrial (mt) DNA regions of the cytochrome b-Cytb, cytochrome c oxidase I-COI16S, and 12S genes→cytochrome b-Cytb, cytochrome c oxidase I-COI16S, and 12S genes of the mitochondrial (mt) DNA   DONE

  1. 55: NAD2? We better explained that we focus on NAD2 in this manuscript following a previous study on NAD5.
  2. 56: for Sparidae→for species of the family Sparidae DONE
  3. 57: fragments→regions DONE
  4. 59: mitochondria→mtDNA DONE
  5. 60: Sparidae family→family Sparidae DONE
  6. 61: mitochondrial→mtDNA DONE

       -   Materials and Methods

  1. 105: MtDNA DONE
  2. 115: 16 sparid species

        -   Results

140: spell out P. erythrinus DONE

p of p-genetic distance is in italic or not? Table and Figure caption

It is in italic, this was checked and corrected across the paper

Spell out all generic name. DONE

  1. 198: What is Ct? This seems to show up first here, but no explanation can be seen anywhere except for the caption of Fig. 8. We specified that Ct is the “Cycle Threshold”

      -    Discussion

  1. 223, 230, 236: sparid DONE
  2. 249: sparid species in the family Sparidae DONE
  3. 257: P. erythrinus DONE

Figure 2. other sparid species DONE

Reviewer 2 Report

COMMENTS TO AUTHORS:

The manuscript “A rapid method for use in end-point and real-time PCR for identification of fish fillets from Pagellus erythrinus and common substitute species against frauds” by Ceruso et al. develops a beautiful, fast and easy molecular tool using NAD2 mtDNA gene (and avoiding other common mtDNA markers such as Cytb, COI, 16S…) for P. erythrinus identification and discrimination from other Sparid and fish species.

In my opinion, the manuscript is well-written and the data has been well analysed and discussed. Therefore, it can represent an interesting contribution for “Foods” readers. However, I have some concerns in its present form.

All my comments and suggestions are enclosed below. I hope they will be welcomed and useful for the authors.

MAJOR ISSUES

  1. Number of individuals analysed: Despite specimens were collected across natural P. erythrinus distribution. Are ten individuals representative of the species? Include on Table 1 that the first eight individuals were collected in FAO 37 area.
  2. Sparid species used: Authors performed a good bioinfomatic job, retrieving mtDNA genomes from NCBI database. However, it is surprising that the gilt-head seabream (Sparus aurata), one of the most important commercially important sparid, was not included on the study when its mtDNA genome is also available (GenBank Accession Number: NC_024236.1). What was the reason for its exclusion?

MINOR ISSUES

- Line 39: Remove last dot from “197.00.”. What is the reason for the decrease in aquaculture production?

- Line 41: Use always two decimals. Replace “1.647” by “1.65”.

- Lines 80-81: Authors state that “the evaluation of primer specificity was extended to other 25 fish species, as reported in Table 2". However there are 26 fish species on Table 2. Replace “Psetta maxima” by “Scophthalmus maximus” for turbot scientific name. Finally, include the family for each species also on Table 2.

- Line 100: Replace “w” by “was”.

-Lines 115-116: Include reference for BioEdit software.

- Lines 141-143: All scientific names must be written in italic letters. Revise throughout the manuscript.

- Lines 147-149: Revise letter size and paragraph style.

-Line 150: Replace “0.2” by “0.20”.

-Line 168: Replace “P. erythrinus” by “P. erythrinus”. Replace “Figure 6,” by “Figure 6”.

- Table 5: Remove decimals from nucleotide counts.

-Line 176: Replace “P. erythrinus” by “P. erythrinus”.

-Line 183: Replace “P. erythrinus” by “P. erythrinus”.

- Line 189: Replace “primer’s” by “primers’”.

-Line 198: Replace “P. erythrinus” by “P. erythrinus”.

-Line 232: Replace “,” by “.” for decimals.

- Line 251: Replace “mt” by “mtDNA”.

-Line 257: Replace “Pagellus erythrinus” by “P. erythrinus”.

Author Response

MAJOR ISSUES

  1. Number of individuals analysed: Despite specimens were collected across natural P. erythrinus distribution. Are ten individuals representative of the species? Include on Table 1 that the first eight individuals were collected in FAO 37 area.

According to the literature, the value of intraspecific genetic variability of sparids is about 0.5-1% (Gyllensten 1985; Faith et al., 2004; Ceruso et al., 2019). Our results on ten specimens were in accordance with similar studies. The value of intraspecific genetic variability of the selected NAD2 fragment was 0% in specimens from the FAO 37 area, and ranged between 0% and 0.8 %, with three nucleotide changes found in the two specimens from Atlantic Ocean.

Our study further showed that NAD2 gene has a very low intraspecific variability degree compared to other genes.

“FAO 37 area” was added to the first eight individuals.

  1. Sparid species used: Authors performed a good bioinfomatic job, retrieving mtDNA genomes from NCBI database. However, it is surprising that the gilt-head seabream (Sparus aurata), one of the most important commercially important sparid, was not included on the study when its mtDNA genome is also available (GenBank Accession Number: NC_024236.1). What was the reason for its exclusion?

We included in all the bioinformatic analysis the species Sparus aurata, GenBank Accession Number: LK_022698, as reported in the Table 4 (with the sparid species considered in this study). Before this analysis, we confirmed that the two complete mtDNA of Sparus aurata (NC_024236.1 and LK_022698) have the same nucleotide sequence.

MINOR ISSUES

- Line 39: Remove last dot from “197.00.”. What is the reason for the decrease in aquaculture production?

The dot was removed.

The potential use of Pagellus erythrinus in aquaculture was investigated in different studies, but it seems that further research on the reproduction and the larval rearing of the species is needed (Klaoudatos et al, 2004; Güner et al., 2004; Micale et al., 2006).

- Line 41: Use always two decimals. Replace “1.647” by “1.65”. DONE

- Lines 80-81: Authors state that “the evaluation of primer specificity was extended to other 25 fish species, as reported in Table 2". However there are 26 fish species on Table 2. 25 was changed in 26

Replace “Psetta maxima” by “Scophthalmus maximus” for turbot scientific name. Finally, include the family for each species also on Table 2. DONE

- Line 100: Replace “w” by “was”.  DONE

-Lines 115-116: Include reference for BioEdit software. DONE

- Lines 141-143: All scientific names must be written in italic letters. Revise throughout the manuscript. DONE

- Lines 147-149: Revise letter size and paragraph style.  DONE

-Line 150: Replace “0.2” by “0.20”. DONE

-Line 168: Replace “P. erythrinus” by “P. erythrinus”. Replace “Figure 6,” by “Figure 6”. DONE

- Table 5: Remove decimals from nucleotide counts. DONE

-Line 176: Replace “P. erythrinus” by “P. erythrinus”. DONE

-Line 183: Replace “P. erythrinus” by “P. erythrinus”. DONE

- Line 189: Replace “primer’s” by “primers’”. DONE

-Line 198: Replace “P. erythrinus” by “P. erythrinus”. DONE

-Line 232: Replace “,” by “.” for decimals. DONE

- Line 251: Replace “mt” by “mtDNA”. DONE

-Line 257: Replace “Pagellus erythrinus” by “P. erythrinus”. DONE

This manuscript is a resubmission of an earlier submission. The following is a list of the peer review reports and author responses from that submission.